# Human Wharton’s Jelly Mesenchymal Stem Cells Secretome Inhibits Human SARS-CoV-2 and Avian Infectious Bronchitis Coronaviruses

**DOI:** 10.3390/cells11091408

**Published:** 2022-04-21

**Authors:** Mohamed A. A. Hussein, Hosni A. M. Hussein, Ali A. Thabet, Karim M. Selim, Mervat A. Dawood, Ahmed M. El-Adly, Ahmed A. Wardany, Ali Sobhy, Sameh Magdeldin, Aya Osama, Ali M. Anwar, Mohammed Abdel-Wahab, Hussam Askar, Elsayed K. Bakhiet, Serageldeen Sultan, Amgad A. Ezzat, Usama Abdel Raouf, Magdy M. Afifi

**Affiliations:** 1Department of Microbiology, Faculty of Science, Al-Azhar University, Assiut 71524, Egypt; mohamed_ali@azhar.edu.eg (M.A.A.H.); ahmedeladly.ast@azhar.edu.eg (A.M.E.-A.); ahmed_wr2000@azhar.edu.eg (A.A.W.); elsayedbakhiet@azhar.edu.eg (E.K.B.); afifi_magdy@ymail.com (M.M.A.); 2Department of Zoology, Faculty of Science, Al-Azhar University, Assiut 71524, Egypt; athabet@azhar.edu.eg (A.A.T.); mnassar919@azhar.edu.eg (M.A.-W.); hussamaskar@azhar.edu.eg (H.A.); 3Reference Laboratory for Veterinary Quality Control on Poultry Production, Animal Health Research Institute, Agricultural Research Center, Dokki, Giza 12618, Egypt; dr.kareemseleem_87@yahoo.com; 4Clinical Pathology, Mansoura Research Center for Cord Stem Cells (MARC-CSC), Faculty of Medicine, Mansoura University, El Mansoura 35516, Egypt; mervatdaood@yahoo.com; 5Department of Clinical Pathology, Faculty of Medicine, Al-Azhar University, Assiut 71524, Egypt; dralisobhy@azhar.edu.eg; 6Proteomics and Metabolomics Research Program, Basic Research Department, Children’s Cancer Hospital, (CCHE-57357), Cairo 57357, Egypt; sameh.magdeldin@57357.org (S.M.); aya.osama@57357.org (A.O.); ali.anwar@57357.org (A.M.A.); 7Department of Physiology, Faculty of Veterinary Medicine, Suez Canal University, Ismailia 41522, Egypt; 8Department of Microbiology, Virology Division, Faculty of Veterinary Medicine, South Valley University, Qena 83523, Egypt; s.sultan@vet.svu.edu.eg; 9Department of Medical Microbiology and Immunology, Faculty of Medicine, Al-Azhar University, Assiut 71524, Egypt; amgadezzat@azhar.edu.eg; 10Department of Botany and Microbiology, Faculty of Science, Aswan University, Aswan 81528, Egypt; oabdulraouf@yahoo.com

**Keywords:** severe acute respiratory syndrome coronavirus 2 (SARS-CoV-2), infectious bronchitis virus (IBV), Wharton’s jelly stem cells, secretome, coronaviruses

## Abstract

Human SARS-CoV-2 and avian infectious bronchitis virus (IBV) are highly contagious and deadly coronaviruses, causing devastating respiratory diseases in humans and chickens. The lack of effective therapeutics exacerbates the impact of outbreaks associated with SARS-CoV-2 and IBV infections. Thus, novel drugs or therapeutic agents are highly in demand for controlling viral transmission and disease progression. Mesenchymal stem cells (MSC) secreted factors (secretome) are safe and efficient alternatives to stem cells in MSC-based therapies. This study aimed to investigate the antiviral potentials of human Wharton’s jelly MSC secretome (hWJ-MSC-S) against SARS-CoV-2 and IBV infections in vitro and *in ovo*. The half-maximal inhibitory concentrations (IC_50_), cytotoxic concentration (CC_50_), and selective index (SI) values of hWJ-MSC-S were determined using Vero-E6 cells. The virucidal, anti-adsorption, and anti-replication antiviral mechanisms of hWJ-MSC-S were evaluated. The hWJ-MSC-S significantly inhibited infection of SARS-CoV-2 and IBV, without affecting the viability of cells and embryos. Interestingly, hWJ-MSC-S reduced viral infection by >90%, in vitro. The IC_50_ and SI of hWJ-MSC secretome against SARS-CoV-2 were 166.6 and 235.29 µg/mL, respectively, while for IBV, IC_50_ and SI were 439.9 and 89.11 µg/mL, respectively. The virucidal and anti-replication antiviral effects of hWJ-MSC-S were very prominent compared to the anti-adsorption effect. In the *in ovo* model, hWJ-MSC-S reduced IBV titer by >99%. Liquid chromatography-tandem mass spectrometry (LC/MS-MS) analysis of hWJ-MSC-S revealed a significant enrichment of immunomodulatory and antiviral proteins. Collectively, our results not only uncovered the antiviral potency of hWJ-MSC-S against SARS-CoV-2 and IBV, but also described the mechanism by which hWJ-MSC-S inhibits viral infection. These findings indicate that hWJ-MSC-S could be utilized in future pre-clinical and clinical studies to develop effective therapeutic approaches against human COVID-19 and avian IB respiratory diseases.

## 1. Introduction

Severe acute respiratory syndrome coronavirus 2 (SARS-CoV-2) and infectious bronchitis virus (IBV) are highly contagious coronaviruses and leading causes of devastating respiratory tract diseases in humans and chickens, respectively. Both SRAS-CoV-2 and IBV are enveloped viruses, with a positive single-stranded RNA genome, and classified into order *Nidovirales*, family *Coronaviridae*, and sub-family *Orthocoronavirinae.* SARS-CoV-2 belongs to the *Betacoronavirus* genus*,* while IBV belongs to the *Gammacoronavirus* genus [1,2,3]. SARS-CoV-2 is the causative agent of the ongoing coronavirus disease 19 (COVID-19) pandemic and is currently the primary global health concern. It initially emerged in December 2019 in Wuhan, a city in the center of China, as devastating pneumonia with a previously unknown etiological pathogen. Subsequently, SARS-CoV-2 has been identified as the causative agent of COVID-19 and officially classified as a novel human coronavirus [1]. Like SARS-CoV-2, IBV infection is the cause of a devastating upper-respiratory tract disease in chickens, known as infectious bronchitis (IB) disease. The virus was initially isolated from infected poultry in the USA in the 1930s [4,5]. IBV infection in poultry flocks causes a substantial economic loss and affects the meat quality and egg production [6]. There are a variety of vaccines developed against SARS-CoV-2 and IBV. However, the high mutation rate and genetic recombination result in the frequent emergence of new variants of these viruses, which significantly impact the efficiency of the currently available vaccines [3,7]. The emergence of new variants and the lack of effective therapeutic agents for SARS-CoV-2 and IBV infections highlight the need for a proper treatment to control the viral spread and mitigate disease progression.

Mesenchymal stem cells (MSCs) possess tremendous potential as therapeutic agents for various human diseases [8,9,10,11,12]. This potential is not fully harnessed because of the side effects associated with MCS-based therapy (MSCT), which include, but are not limited to, tumorigenesis, immune rejection, and infection [13,14,15,16]. MSCs secrete many biologically active extracellular factors, including soluble proteins, nucleic acids, and extracellular vesicles [17]. These secreted factors are collectively known as stem cells secretome. MSC secretome (MSC-S) modulates the communications between stem cells and surrounding cells [18,19]. Cell-free secretome was able to mimic stem cells’ immunomodulatory and tissue-regenerative abilities [17,18,20]. Previous studies indicated that MSC-S could successfully replace stem cells in various MSCT [21,22]. Using cell-free secretome bypasses MSCT-associated side effects. In addition, secretome has low immunogenicity compared to their cell of origin, so it can aid in developing a disease-specific universal therapeutic protocol [19,23,24,25,26].

Human Wharton-jelly-derived MSCs (hWJ-MSCs) have recently gained more attention for their superior proliferation rate, immune-privileged characteristics, and lower carcinogenic profile after in vivo transplantation [27,28,29]. hWJ-MSCs have characteristics of embryonic stem cells yet have no ethical concerns. Moreover, it can be easily isolated from the readily available umbilical cord (UC). These advantages grant hWJ-MSCs unique features compared to bone marrow MSCs (BMSCs) and adipose tissue stem cells (ASCs) [30,31]. The study aimed to evaluate the antiviral potential of hWJ-MSCs secretome against SARS-CoV-2 and IBV infections, as well as to profile their protein contents.

## 2. Materials and Method

### 2.1. Cells and Viruses

Vero-E6 (African green monkey kidney) cells were used to study the antiviral effect of hWJ-MSC-S in vitro. Cells were propagated in Dulbecco’s modified Eagle’s medium (DMEM) (GIBCO, Waltham, MA, USA) containing 10% fetal bovine serum (FBS) (GIBCO, Waltham, MA, USA), and 1% antibiotic-antimycotic (AA) mixture (GIBCO, Waltham, MA, USA) at 37 °C in a humidified atmosphere of 5% CO_2_. SARS-CoV-2, hCoV-19/Egypt/NRC-03/2020 (Accession Number on GSAID: EPI_ISL_430820) [32] and IBV, wild type IBV-EGY/CH/CV10-2019 [33], viruses were propagated in Vero-E6 cells and titrated using plaque titration assay [34].

### 2.2. Collection and Processing of Human Umbilical Cords (hUCs)

hUC samples (*n* = 20) were obtained from the department of obstetrics and gynecology, Al-Azhar University Hospital in Assiut. This study was approved by the ethical committees of faculty of medicine and the faculty of science at Al-Azhar University in Assiut (APPROVAL NUMBER/ID:202015). Informed consent was obtained from all individuals who participated in the study. hUCs were collected under sterile conditions after normal full-term healthy pregnancy from 20 healthy mothers and transported to the laboratory in phosphate-buffered saline (PBS) supplemented with antibiotics, 100 U/mL penicillin, and 100 μg/mL streptomycin. At the laboratory, the outer surface of UC was sterilized by 70% ethanol and the cord was washed twice with PBS and serum-free DMEM (GIBCO, USA) to remove excess blood. The UC was cut longitudinally with a sterile surgical scissor and Wharton’s Jelly (WJ) within the cord was scraped using a scalpel. Then, the cord blood vessels (two arteries and one vein) were removed, and the remaining cord tissue (CT) was collected. WJ and CT were minced separately into 1–2 mm pieces before digestion with the enzymatic blend for 30 min at 37 °C in a 5% CO_2_ incubator [35].

### 2.3. Culturing of hUC-Derived Mesenchymal Stem Cells (hUC-MSCs)

Partially digested WJ and CT pieces were plated separately in a six-well plate with DMEM/F12 (GIBCO, Waltham, MA, USA) supplemented with 10% FBS and 1% AA solution. Plates were incubated at 37 °C in a 5% CO_2_ incubator [35]. After 7 days, hUC pieces were removed, and the culture medium was replaced with fresh ones. Cells were grown to reach 80% confluency before passaging. The isolation was considered successful if isolated cells were cultured and maintained up to the 5th passage (P5) without any contamination. Isolated stem cells were cryopreserved in FBS supplemented with 10% DMSO.

### 2.4. Flow Cytometry Characterization of hUC-MSCs

hUC-MSC cells were detached using 0.25% Trypsin-EDTA and collected by centrifugation at 1000× *g* for 10 min. Then, 10^6^ hWJ-MSCs in 100 μL volume were stained with 10 μL of Peridinium-chlorophyll-protein (Per-CP)-conjugated anti-CD105/Endoglin (Mouse IgG1; Clone 166707, R&D Systems, McKinley Place, MN, USA), Carboxyfluorescein (CFS)-conjugated-CD73 (Mouse IgG2B; Clone 606112, R&D Systems, McKinley Place, MN, USA), allophycocyanin (APC) conjugated anti-CD90/Thy1 (Mouse IgG2A; Clone Thy-1A1, R&D Systems, McKinley Place, MN, USA), phycoerythrin (PE) conjugated anti-CD45 (Mouse IgG1; Clone 2D1, R&D Systems, McKinley Place, MN, USA), and PE-CD34 (Mouse IgG1; Clone QBEnd10, R&D Systems, McKinley Place, MN, USA) monoclonal antibodies for 30 min. The viability of cells was monitored using 7-aminoactinomycin D (7-AAD) staining. A total of 50,000 events were acquired and analyzed using the FACS Cantor flow cytometer (Becton Dickinson Biosciences, Franklin Lakes, NJ, USA) and Kaluza analysis software 1.5a (Beckman Coulter, Brea, CA, USA).

### 2.5. Collection of hWJ-MSC-S

hWJ-MSCs were grown in a complete growth medium to reach 80% confluency before replacing the culture medium with a serum-free medium. After 48 h, Conditioned medium (CM) was harvested, centrifuged at 1000× *g* for 10 min to remove cells residues, and stored at −80 °C until used in subsequent experiments.

### 2.6. Cytotoxicity Assay

The 3-(4, 5-dimethylthiazol-2-yl)-2, 5-diphenyltetrazolium bromide (MTT) assay was performed to evaluate the cytotoxicity of hWJ-MSC-S in Vero-E6 cells as previously described [36]. Briefly, cells were seeded in a 96-well plate in DMEM containing FBS (10%), and AA solution (1%). After 24 h, the growth medium was aspirated, and cells were washed twice with 1X PBS. Then, cells were treated with different concentrations of hWJ-MSC-CM. At 24 h post-treatment, the medium was removed, cells were washed with 1x PBS, MTT solution (20 μL/well of stock solution) was added, and cells were incubated for 4 h. The absorbance was measured at 450 nm. The percentage of cytotoxicity compared to the untreated cells was determined with the following equation: $$\%\;\mathrm{cytotoxicity}=\frac{\left(\mathrm{absorbance}\;\mathrm{of}\;\mathrm{cells}\;\mathrm{without}\;\mathrm{treatment}-\mathrm{absorbance}\;\mathrm{of}\;\mathrm{cells}\;\mathrm{with}\;\mathrm{treatment}\right)\times 100}{\mathrm{absorbance}\;\mathrm{of}\;\mathrm{cells}\;\mathrm{without}\;\mathrm{treatment}\;}$$

The plot of % cytotoxicity versus sample concentration was used to calculate the concentration which exhibited 50% cytotoxicity (CC50).

### 2.7. Embryotoxicity

Embryotoxicity of hWJ-MSC-S was evaluated using toxicity assay *in ovo* as previously described [37,38]. Briefly, different concentrations of hWJ-MSC-S were inoculated in 0.2 mL volume into the allantoic cavity of 10-day-old Specific pathogen-free embryonated chicken eggs (SPF-ECEs) (10 eggs per concentration). The SPF-ECEs were incubated at 37 °C for 5 days and inspected daily by candling to check the embryo viability. Eggs inoculated by serum-free DMEM or left without inoculation were used as negative and blank controls, respectively.

### 2.8. Determination of Viral Inhibitory Concentration 50 (IC_50_)

The IC50 value of hWJ-MSC-S was determined as previously described [39,40,41]. Briefly, Vero-E6 cells were seeded at a concentration of 2.4 × 10^4^ in 96-well plates and incubated overnight at 37 °C and 5% CO_2_ condition. Serial dilutions of hWJ-MSC-S were mixed with 10^3^ PFU of virus (SRAS-CoV-2 or IBV) and incubated at room temperature (RT) for 1 h. Cell monolayers were washed with PBS and inoculated with hWJ-MSC-CM/virus mixtures, and incubated at 37 °C and 5% CO_2_ for 72 h. Next, cells were fixed with 100 μL of 4% paraformaldehyde for 20 min and stained with 0.1% crystal violet in distilled water for 15 min at RT. The crystal violet dye was then dissolved using 100 μL absolute methanol per well, and the optical density was measured at 570 nm using Anthos Zenyth 200 rt plate reader (Anthos Labtec Instruments, Heerhugowaard, The Netherlands) [36]. The IC_50_ of the hWJ-MSC-S is that required to reduce the virus-induced cytopathic effect (CPE) by 50%, compared to the untreated virus control. The infection assay was performed in BSL-3 facility.

### 2.9. Plaque-Reduction Assay

Plaque-reduction assay was used to determine the antiviral activity of hWJ-MSC-S as previously described [42]. Briefly, Vero-E6 cells were seeded at a concentration of 1 × 10^5^ in a 6-well plate and incubated overnight at 37 °C. Following this, 200 µL (10^3^ PFU) of the virus was mixed with different non-toxic concentrations of hWJ-MSC-S and incubated at RT for 1 h before inoculation onto the monolayer of Vero-E6 cells. One hour later, the supernatant was aspirated, cells were washed, overlaid with DMEM medium supplemented with 2% agarose, 1% AA mixture, and 4% bovine serum albumin (BSA), and incubated at 37 °C. Three days later, the overlay medium was discarded, and cells were fixed for 1 h in 10% formalin solution and stained with 0.1% crystal violet working solution. The infection experiment was performed in BSL-3 facility. The plaque-forming units (PFUs) were counted, and percentages of reduction were calculated according to the following equation:$$\begin{array}{c} \%\;\mathrm{Viral}\;\mathrm{inhibition}\\ =\frac{\left(\mathrm{PFU}\;\mathrm{number}\;\mathrm{of}\;\mathrm{virus}\;\mathrm{control}\;-\;\mathrm{PFU}\;\mathrm{number}\;\mathrm{after}\;\mathrm{the}\;\mathrm{treatment}\;\mathrm{with}\;\mathrm{hWJ}-\mathrm{MSC}-\mathrm{CM}\right)\;\times \;100}{\mathrm{PFU}\;\mathrm{number}\;\mathrm{of}\;\mathrm{virus}\;\mathrm{control}\;} \end{array}$$

### 2.10. Determination of the Mode of Action

The mode of antiviral activity of hWJ-MSC-S was determined by evaluation of its virucidal, anti-adsorption, and anti-replication effects in Vero-E6 cells as previously described [36,43]. To evaluate the virucidal activity, 200 µL (10^3^ PFU) of the virus was mixed with different non-toxic concentrations of hWJ-MSC-S and incubated at RT for 1 h before infecting the cell monolayer for 1 h at 37 °C. To determine the anti-adsorption effect, the cell monolayer was pre-treated with different concentrations of hWJ-MSC-S for 2 h at 4 °C before infection with 200 µL (10^3^ PFU) of the virus for 1 h at 37 °C. In another set of experiments, the cell monolayer was infected with 200 µL (10^3^ PFU) of the virus at first for 1 h at 37 °C and then treated with hWJ-MSC-S for 1 h at 37 °C to determine the anti-replication effect. In all protocols, the supernatant was aspirated, cells were washed, and incubated with DMEM medium supplemented with 2% agarose, 1% AA mixture, and 4% BSA at 37 °C. Three days later, the overlay medium was discarded, and cells were fixed for 1 h in 10% formalin solution and stained with 0.1% crystal violet working solution. The PFUs were counted, and the percentages of reduction were calculated as described above.

### 2.11. In Ovo Anti-IBV Activity of hWJ-MSC-S

IBV (10^5^ EID_50_/0.1 mL) was mixed with 0.1 mL of different non-toxic concentrations of hWJ-MSC-S and incubated at RT for 1 h before inoculation into the allantoic cavity of 10-day-old SPF-ECEs (five eggs per concentration). Inoculated SPF-ECEs were incubated at 37 °C for 5 days and inspected daily by candling to check the embryo viability [44]. SPF-ECEs inoculated by DMEM or left without inoculation were used as a negative and blank control, respectively. Three days post-inoculation, allantoic fluid was collected and subjected to RNA extraction and reverse transcription-quantitative real-time polymerase chain reaction (RT-qPCR) to detect IBV copies.

### 2.12. RNA Extraction and RT-qPCR to Detect IBV

Viral RNA was extracted from allantoic fluids using the QIAamp Viral RNA mini kit (Qiagen, Hilden, Germany) as per the manufacturer’s recommendations. The RNA concentration was measured with a NanoDrop ND-2000 spectrophotometer (Thermo Fisher Scientific< Waltham, MA, USA). The forward primer IBV (IBV-F: 5′-ATGCTCAACCTTGTCCCTAGCA-3′), reverse primer (IBV-R: 5′-TCAA-ACTGCGGATCA-TCACGT-3′), and TaqMan^®^ probe (IBV-TM: FAM-TTGGAAGTAGAGTGACGCC-CAAACTTCA-BHQ1) specific to IBV N gene were used to determine IBV copy numbers in one step RT-qPCR [7]. The master mix was utilized in a total volume of 25 µL containing 12.5 µL of QuantiTect RT-PCR kit (Qiagen, Hilden, Germany), 0.5 µL of each primer (50 pmol), 0.125 µL (30 pmol) of the probe, 0.25 µL of RT-enzyme, 8.125 µL of RNase-free water and 3 µL of RNA template [45]. The reaction was performed and analyzed using a Stratagene MX3005P real-time PCR machine (Agilent technologies, Santa Clara, CA, USA).

### 2.13. Mass Spectrometry Analysis of hWJ-MSC-CM

Proteins were precipitated from hWJ-MSC-S samples collected with four-times chilled Acetone. After incubation at −80 °C for 30 min and at −20 °C overnight, samples were centrifuged at 10,000 rpm for 30 min. The protein extract of hWJ-MSC-S was denaturated by placing 50 μL lysis solution (8 M urea, 500 mM Tris HCl, pH 8.5) with complete ultra-proteases (Roche, Mannheim, Germany). Samples were incubated at 37 °C for 1 h with occasional vortex, and then centrifuged at 12,000 rpm for 20 min. The protein concentration of hWJ-MSC-S was assayed using the bicinchoninic acid (BCA) method (Pierce, Rockford, IL, USA) at Å562 nm before digestion. Next, 30 µg of hWJ-MSC-S protein was subjected to in-solution digestion. In brief, protein pellets were re-suspended in an 8 M urea lysis solution and reduced with 5 mM tris 2-carboxyethyl phosphine (TCEP) for 30 min. The alkylation of cysteine residues was performed using 10 mM iodoacetamide for 30 min in a dark area. Samples were diluted to a final concentration of 2 M urea with 100 mM Tris-HCl, pH 8.5, before digestion with trypsin [46]. For endopeptidase digestion, modified procaine trypsin (Sigma, Darmstadt, Germany) was added at 30: 1 (protein: protease mass ratio) and incubated overnight in a thermo-shaker at 600 rpm at 37 °C. The digested peptide solution was acidified using 90% formic acid to a final pH of 2.0. The resultant peptide mixture herein was cleaned up using the stage tip as discussed earlier [46]. Each sample was run in triplicates. Nano-LC MS/MS analysis was performed using Triple TOF 5600 + (AB Sciex, Darmstadt, Canada) interfaced at the front end with Eksigent nanoLC 400 autosamplers with Ekspert nanoLC 425 pump. On trap and elute mode, peptides were trapped on CHROMXP C18CL 5 μm (10 × 0.5 mm) (AB Sciex, Darmstadt, Germany). MS and MS/MS ranges were 400–1250 *m/z* and 170–1500 *m/z*, respectively. A design of 120-min linear gradient 3–80% solution (80% ACN, 0.2% formic acid) was used. The 40 most intense ions were sequentially selected under data-dependent acquisition (DDA) mode with a charge state 2–5. For each cycle, survey full scan MS and MS/MS spectra were acquired at a resolution of 35.000 and 15.000, respectively. To ensure accuracy, external calibration was scheduled and run during sample batches to correct possible TOF deviation.

### 2.14. Bioinformatics Analysis

Mascot generic format (mgf) files were generated from the raw file using the script supplied by AB Sciex. MS/MS spectra were searched using X Tandem in Peptide shaker (version 1.16.26) against Uniprot *Homosapiens* (Swiss-prot and TrEMBL database containing 224,139 proteins) with target and decoy sequences. The search space included all fully and semi-tryptic peptide candidates with a maximum of 2 missed cleavages. Precursor mass and fragment mass were identified with an initial mass tolerance of 20 ppm and 10 ppm, respectively. The carbamidomethylation of cysteine (+57.02146 amu) was considered as a static modification and oxidation at methionine (+15.995), acetylation of protein N- terminal and K (+42.01 amu), and pyrrolidone from carbamidomethylated C (−17.03 amu) as variable modification. To assure a high-quality result, the false discovery rate (FDR) was kept at 1% at the protein level. Final assembly and emerging of sample replicates were applied to generate the final outputs of each sample using in-house software ‘ProteoSelector’ (https://www.57357.org/en/department/proteomics-unit-dept/in-house-bioinformatics-tools/ (accessed on 3 December 2021)) (Appendix A). Normalized spectral abundance factor (NSAF) was averaged for each protein [47,48].

### 2.15. Gene Ontology and Pathway Enrichment Analysis

Gene ontology was performed on uniquely retrieved genes from the consolidated proteome profile of hWJ-MSC-S and searched against gene ontology database biological process, molecular function, cellular component, and protein class analysis was utilized with Fisher’s Exact test with significance level 0.05 and 5% FDR. Or pathway enrichment analysis, Enrichr’s web-based tool was used for pathway enrichment analysis with a COVID-19 database search [49,50]. The retrieved result was filtered at an adjusted *p*-value of 0.05 and an odds ratio of more than 1.

### 2.16. Statistical Analysis

Data were shown as means ± standard deviation (SD) of three independent experiments. One-way ANOVA and Student’s *t*-test were used to compare different treatments and quantify significance using Prism (5.0; GraphPad Software, La Jolla, CA, USA). *p* values < 0.05 were considered statistically significant. Graphics were drawn by R [51] and Prism.

## 3. Results

### 3.1. Isolation and Characterization of hUCMSC

The overall isolation of UC-MSCs was successful in 15 out of 20 (75%) of the collected umbilical cords. Cells were successfully isolated from 12 out of 15 (80%) and 6 out of 15 (40%) of the collected WJ and CT samples, respectively. The isolation rate was significantly higher in WJ than in CT, as shown in Figure 1A. Isolated cells showed normal spindle-shaped stem cells with adherent properties (Figure 1B). There were no cell morphology differences between WJ- and CT-derived stem cells. Flow cytometry characterization of isolated UC-MSCs showed that they were positive for the expression of MSC markers CD73, CD105, and CD90, whereas they were negative for hematopoietic stem cell markers CD45 and CD34 (Figure 1C). hWJ-MSCs were further maintained for 48 h to collect their secretome, as illustrated in Figure 1D.

### 3.2. Cytotoxicity and Antiviral Activity of hWJ-MSC-S

The cytotoxicity and antiviral activity of hWJ-MSC-S against SARS-CoV-2 and IBV were evaluated using the Vero-E6 cell line. The half-maximal inhibitory concentration (IC_50_), half-maximal cytotoxicity concentration (CC_50_), and selective index (SI) of hWJ-MSC-S were calculated. To determine CC_50_, Vero-E6 cells were treated with different concentrations of hWJ-MSC-S, and the cytotoxicity levels were measured using an MTT assay. The CC50 value of hWJ-MSC-S was 39,200 µg/mL (Figure 2A). To evaluate the antiviral effect of hWJ-MSC-CM, SARS-CoV-2 and IBV were individually treated with multiple non-toxic concentrations of hWJ-MSC-S before infecting Vero-E6 cells. The antiviral activity of hWJ-MSC-S was determined using a cytopathic effect assay, and plaque-reduction assay. hWJ-MSC-S significantly reduced the cytopathic effect induced by SARS-CoV-2 and IBV infection in Vero-E6 (Figure 2B). IC50 values of hWJ-MSC-S against SARS-CoV-2 and IBV were 166.6 µg/mL and 439.9 µg/mL, respectively (Figure 2B). In addition, the SI (CC50/IC50) value of hWJ-MSC-S against SARS-CoV-2 and IBV was 235.29 and 89.11, respectively (Table 1). Similarly, the plaque reduction assay showed that plaque formed by SARS-CoV-2 and IBV was reduced by hWJ-MSC-S in a dose-dependent manner (Figure 2C). hWJ-MSC-S reduced PFU/mL of SARS-CoV-2 and IBV by >90%, at a concentration of 1000 µg/mL (Figure 2C), while at a concentration of 250 µg/mL of hWJ-MSC-S, there was a variation in the viral inhibition between SARS-CoV-2 and IBV (Figure 2C). At a 125 µg/mL concentration, the antiviral activities of hWJ-MSC-S against both SARS-CoV-2 and IBV were significantly decreased.

### 3.3. Determination of the Antiviral Mechanism of hWJ-MSC-S

The hWJ-MSC-S mode of action was determined by investigating the antiviral activity of hWJ-MSC-S against SARS-CoV-2 and IBV infection in Vero-E6 cells, using three different antiviral protocols: (i) virucidal, (ii) inhibition of viral adsorption, and (iii) inhibition of viral replication, as illustrated in Figure 3A. Although hWJ-MSC-S inhibited SARS-CoV-2 infection in all antiviral protocols, there was a significant variation in the percentage of inhibition among different antiviral assays and hWJ-MSC-S concentrations (Figure 3B,C, Appendix A). At a 1000 µg/mL concentration, hWJ-MSC-S inhibited SARS-CoV-2 infection by >95%, 85%, and 42% in virucidal, anti-replication, and anti-adsorption protocols, respectively (Figure 3B and Appendix A), whereas 500 µg/mL and 250 µg/mL concentrations of hWJ-MSC-S inhibited viral infection by >70%, >60%, and <50% in virucidal, anti-replication, and anti-adsorption protocols, respectively. These results indicated that the hWJ-MSC-S inhibited SARS-CoV-2 infection directly by inactivating the virion and indirectly by inhibiting viral replication. Interestingly, hWJ-MSC-S inhibited IBV infection similarly to SARS-CoV-2. At 1000 µg/mL concentration, hWJ-MSC-S inhibited IBV infection by >90% using virucidal and anti-replication protocols, and >50% inhibition was observed in the anti-adsorption protocol (Figure 3C and Appendix A), while at 500 µg/mL concentration of hWJ-MSC-S, viral inhibition was >60% in virucidal and anti-replication protocols, and <40% in the anti-adsorption protocol. At 250 and 125 µg/mL concentrations, the antiviral effect of hWJ-MSC-S against SARS-CoV-2 and IBV was significantly reduced (Figure 3B,C, Appendix A).

### 3.4. Determination of in Ovo Anti-IBV Activity of hWJ-MSC-S

The embryotoxicity and antiviral activity of hWJ-MSC-S against IBV were evaluated using 10-day-old SPF-ECEs. The lethal dose 50 (LD_50_) was assessed by inoculating SPF-ECEs with various hWJ-MSC-S concentrations and embryo viability was checked daily. The LD_50_ was calculated as the concentration that causes the death of 50% of inoculated embryos. The LD_50_ value of hWJ-MSC-S was 4121 µg/mL (Figure 4A). To evaluate the inhibitory effect of hWJ-MSC-S against IBV in the *in ovo* model, IBV was treated with different non-toxic concentrations of hWJ-MSC-S for 1 h before inoculation into the allantoic cavity of SPF-ECEs, as illustrated in Figure 4B. After 72 h, the allantoic fluid was collected from all groups and subjected to RT-qPCR to evaluate the reduction in IBV titer. hWJ-MSC-S significantly reduced the virus titer (Figure 4C). Treating IBV with hWJ-MSC-S at a 1000 µg/mL concentration significantly reduced the viral titer in the allantoic fluid of SPF-ECEs. Viral titer was reduced from 763,000 ± 32,638 EID_50_/_mL_ to 38.333 ± 6.0093 EID_50_/_mL_ (Figure 4C). At a concentration of 500 µg/mL, hWJ-MSC-S reduced IBV titer from 763,000 ± 32,638 EID_50_/_mL_ to 1896 ± 926.1 EID_50_/_mL_. In addition, 250 µg/mL of hWJ-MSC-S reduced IBV titer from 763,000 ± 32,638 EID_50_/_mL_ to 20,380 ± 2885 EID_50_/_mL_, while 125 µg/mL of hWJ-MSC-S reduced IBV titer from 763,000 ± 32,638 EID_50_/_mL_ to 437,700 ± 18,480 EID_50_/_mL_ (Figure 4C). Percentages of IBV inhibition were >99%, >97%, >96% at hWJ-MSC-S concentrations of 1000 µg/mL, 500 µg/mL, and 250 µg/mL, respectively (Figure 4D), whereas, at a 125 µg/mL concentration, the viral inhibition by hWJ-MSC-S significantly dropped to 40% (Figure 4D).

### 3.5. Proteomics Profiling and Gene Ontology of hWJ-MSC-S

Proteomics profiling of hWJ-MSC-S was run in triplicate. NSAF metrics represent protein abundance. Figure 5A and Appendix A show NSAF abundancies of the top 50 proteins, clustered using the Ward algorithm [52]. A complete proteomics profile was obtained by merging the three replicates. As a result, 259 proteins mapped to 252 genes were listed. Additional data could be found in Appendix A. Next, we parsed the gene ontology of the proteome profile. Interestingly, results showed significant enriched biological processes involved in immune response activation, namely: Annexin A1, Heat shock protein 90, and collagen alpha 1. We also reported a significant opsonization component, including complement C3, Pentraxin-related protein, and Spondin. Regulation of leukocyte migration was overexpressed with Thrombospondin-1, Plasminogen activator inhibitor 1, and extracellular matrix protein 1 (Figure 5B and Appendix A). Protein class analysis showed significant fractions involved in defense immune proteins (Figure 6A). Finally, pathway enrichment analysis showed significantly activated pathways, characteristic of COVID-19 (Figure 6B and Appendix A).

## 4. Discussion

SARS-CoV-2 and IBV are highly infectious coronaviruses and leading causes of morbidity and mortality rates in infected hosts [53,54,55]. Currently, there are no effective methods or proven therapies to control the viral spread and mitigate disease progression for SARS-CoV-2 and IBV infections [56,57,58]. Furthermore, inherited mutagenic characteristics of SARS-CoV-2 and IBV limit the efficacy of available vaccines and exacerbate the disease situations [59,60]. Thus, the development of effective antiviral therapeutic approaches for SARS-CoV-2 and IBV infections is a critical step toward controlling COVID-19 and IB diseases. Secreted factors of stem cells are gaining more attention as alternatives to stem cells to avoid the associated side effects. Exploiting stem cell secretome offers a variety of advantages, including, but not limited to, improved safety and efficacy, flexibility in the storage and handling, and significantly enhanced immune tolerance [61,62,63,64]. The antiviral potentials of hWJ-MSC-S against SARS-CoV-2 and IBV are mainly unexplored.

In this study, we investigated the antiviral activity of hWJ-MSC-S against SARS-CoV-2 and IBV infections. Firstly, we isolated and characterized Wharton’s Jelly mesenchymal stem cells and collected their secretome. Secondly, we demonstrated the antiviral inhibitory effects of hWJ-MSC-S on SARS-CoV-2 and IBV infections. Thirdly, we showed that hWJ-MSC-S inhibits viral infection, mainly through the inactivation of virus particles and the inhibition of viral replication. Fourthly, *in ovo* antiviral activity of hWJ-MSC-S against IBV was demonstrated by its ability to decrease viral titers and to protect chicken embryos. Fifthly, we identified the protein composition of hWJ-MSC-s and pinpointed the major immunomodulatory key players and significant pathways involved.

In this study, the isolation rate of mesenchymal stem cells from Wharton’s Jelly was more efficient than cord lining [35,65,66]. This could be attributed to hWJ-MSC’s high proliferation and growth rates and highlights their therapeutic potential [67,68]. hWJ-MSC-S exhibited very low cytotoxicity and embryotoxicity levels and its CC_50_ and LD_50_ values were very high. Accordingly, this result confirms the previous findings that have proven the safety and efficacy of stem-cell-secreted factors [62]. hWJ-MSC-S significantly inhibited SARS-CoV-2 and IBV infections in a dose-dependent manner.

Interestingly, hWJ-MSC-S reduced plaque formed by SARS-CoV-2 and IBV in Vero-E6 cells by more than 90%. Similarly, in the *in ovo* model, hWJ-MSC-S significantly reduced viral titer and enhanced embryo survival compared to the control (*p* < 0.0001). Although hWJ-MSC-S similarly inhibited SARS-CoV-2 and IBV infections, there were variations in the inhibitory concentrations between the two viruses. It showed markedly higher IC_50_ and SI against SARS-CoV-2 in comparison to IBV. In addition to that, IBV was less sensitive to inactivation by hWJ-MSC-S. This suggests that hWJ-MSC-S may exploit different strategies to inhibit SARS-CoV-2 and IBV infections. Further, the differences in sensitivity to hWJ-MSC-S between the two viruses could be attributed to the nature of each virion. Moreover, IBV is relatively resistant to inactivation with soluble antiviral factors compared to other coronaviruses, such as SARS-CoV [69,70].

The hWJ-MSC-S inhibited SARS-CoV-2 and IBV infections using different strategies—directly by virucidal effect and indirectly via inhibiting viral replication. This variation in the mode of action could be ascribed to the wide range of molecules that have been identified in hWJ-MSC-S proteomic analysis (Figure 5 and Figure 6). In general, stem cell secretome contains various bioactive factors, including paracrine molecules (such as growth factors, cytokines, and microRNAs(miRNAs)) and extracellular vesicles (microvesicles and exosomes). These bioactive molecules modulate the activities of stem cell secretome [71,72]. Our proteomic data revealed a significant enrichment of immunomodulatory and antiviral proteins, including Thymosin beta-4 (T*β*4), Peptidyl-prolyl cis-trans isomerase A (PPIase A), and tissue metalloproteinase inhibitors 1 (TIMP1) in hWJ-MSC-S. T*β*4 is an anti-inflammatory and antioxidant protein that has been shown to inhibit coronavirus mouse hepatitis virus (CoV MHV) Infection in vivo [73]. PPIase A, also known as cyclophilin A (CypA), is an enzyme that catalyzes the cis-trans isomerization of proline imidic peptide bonds in oligopeptides [74,75]. The TIMP1 is a zinc-dependent enzyme known to inhibit MMP activity [76]. Mesenchymal stem cells (MSCs) are known to secrete high levels of TIMPs [77].

T*β*4 has been reported to significantly increase the survival of CoV-MHV-A59-infected mice by inhibiting viral replication and alleviating infection-associated immune activation and liver damage [73]. In addition, levels of serum T*β*4 have been associated with hepatitis B virus-related liver failure, and higher T*β*4 values indicated improvement and recovery from the disease [78]. Interestingly, thymosin treatment has been used recently to alleviate the severity of COVID-19 symptoms [79]. Like T*β*4, CypA is involved in the life cycles of several viruses, including SARS-CoV, CoV-229E, CoV-NL63, FCoV, HIV-1, HCV, and influenza virus [74,75,80,81]. The CypA has been shown to impair influenza virus replication through degradation of the viral M1 protein [75,82]. Recently, CypA and TIMP1 have been suggested as potential treatments for SARS-CoV-2 infection and COVID-19 [83,84,85,86,87].

Like stem cell-derived soluble factors, extracellular vesicles (EVs) in stem cell secretome have been shown to carry out immunomodulatory and antiviral activity [88,89,90,91]. UC-MSC-derived exosomes inhibited SRAS-CoV-2 infection and associated lung inflammation [92,93,94]. Furthermore, exosomes competitively blocked SARS-CoV-2 entry via its surface-associated human angiotensin-converting enzyme 2 (hACE2) [95]. Given the well-documented immunomodulatory and antiviral activities of stem cell soluble factors and EV, the antiviral effects of hWJ-MSC-S on SARS-CoV-2 and IBV may be mediated by its soluble factors (including T*β*4 and CypA and TIMP1) and EVs (MVs and EXos). These findings imply that hWJ-MSC-S-based therapeutic agents can potentially be used in future studies to target SARS-CoV-2 and IBV infections and their associated diseases. The low cytotoxicity and embryotoxicity profiles of hWJ-MSC-S indicated that it could be used safely *in vivo* studies, particularly to treat IBV-infected poultry flocks. Although drinking water and feed medication are common practices for the treatment of poultry diseases, for hWJ-MSC-S-based anti-IBV therapies, the parenteral administration would be the main route. This is because the biologically active extracellular paracrine factors of hWJ-MSC-S could be deactivated by impurities in drinking water and feed additives. Moreover, a lyophilized form of hWJ-MSC-S could be more suitable for drug administration to poultry flocks. Further, hWJ-MSC-S could be used therapeutically to treat IBV-infected chickens, as well as prophylaxis to control IBV infection in chicken flocks.

## 5. Conclusions

In this work, we have demonstrated the antiviral activity of hWJ-MSC-S against two medically important coronaviruses: human SARS-CoV-2 and avian IBV. The hWJ-MSC-S significantly inhibited SARS-CoV-2 and IBV infection in vitro and *in ovo,* respectively. Our antiviral mechanistic studies suggest that the antiviral effects of hWJ-MSC-S are mainly mediated via virucidal and anti-replication mechanisms. Using proteome profiling and gene ontology, we have identified various immunomodulatory and antiviral bioactive factors in hWJ-MSC-S. This study provides a perspective on new antiviral therapeutics against SARS-CoV-2 and IBV infections, using hWJ-MSC-S-based approaches. Findings from this study may, thus, have implications for developing effective strategies to combat COVID-19, IB, and future coronavirus outbreaks. While this study aimed to evaluate the antiviral effect of hWJ-MSC-secreted factors collectively, it will be particularly meaningful to determine the role of an individual secreted factor in the hWJ-MSC-S-mediated antiviral effect.

## Figures and Tables

**Figure 1 cells-11-01408-f001:**
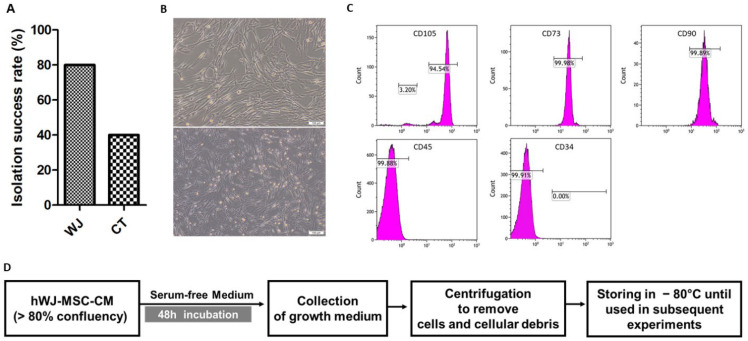
Isolation and characterization of hUCMSCs. (**A**) Percentage of the isolation success rate of hUCMSCs. (**B**) Morphology of hWJ-MSCs was observed under light microscopy at 200× magnification. (**C**) Flow cytometry characterization of hWJ-MSCs demonstrates positive expression of mesenchymal stem cells markers: CD105, CD73, CD90, and negative expression of hematopoietic stem cells markers: CD45, and CD34. (**D**) Schematic depicting the collection of hWJ-MSCs secretome.

**Figure 2 cells-11-01408-f002:**
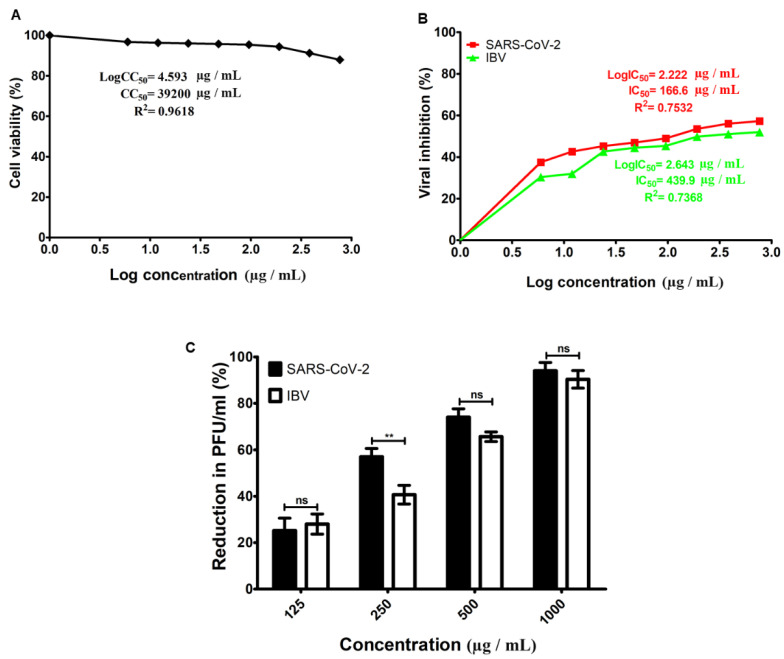
Cytotoxicity and antiviral activity of hWJ-MSC-S. (**A**) Cytotoxicity concentration (CC_50_) of hWJ-MSC-S on Vero-E6 cells. Cells were treated with different concentrations of hWJ-MSC-S for 24 h. The cytotoxicity levels were measured using an MTT assay. (**B**) Half-maximal inhibitory concentration (IC_50_) of hWJ-MSC-S against SARS-CoV-2 infection (red line) and IBV infection (green line) in Vero-E6 cells. Virus incubated with different concentrations of hWJ-MSC-S for 1 h before infecting Vero-E6 cells. The IC_50_ was calculated as the concentration of hWJ-MSC-S that was required to reduce the virus-induced cytopathic effect (CPE) by 50%, compared to the virus control. (**C**) Reduction in plaque formation after treatment of SARS-CoV-2 and IBV with different concentrations of hWJ-MSC-S (125, 2,505,001,000 µg/mL). IC_50_ and CC_50_ values were calculated using nonlinear regression analysis by plotting log inhibitor versus normalized response (Variable slope). Results are shown as means ± SD of three independent experiments, each run in triplicate. The *p* value ** *p* < 0.01 indicates the significant correlation between the antiviral activities of hWJ-MSC-S against SARS-CoV-2 vs IBV.

**Figure 3 cells-11-01408-f003:**
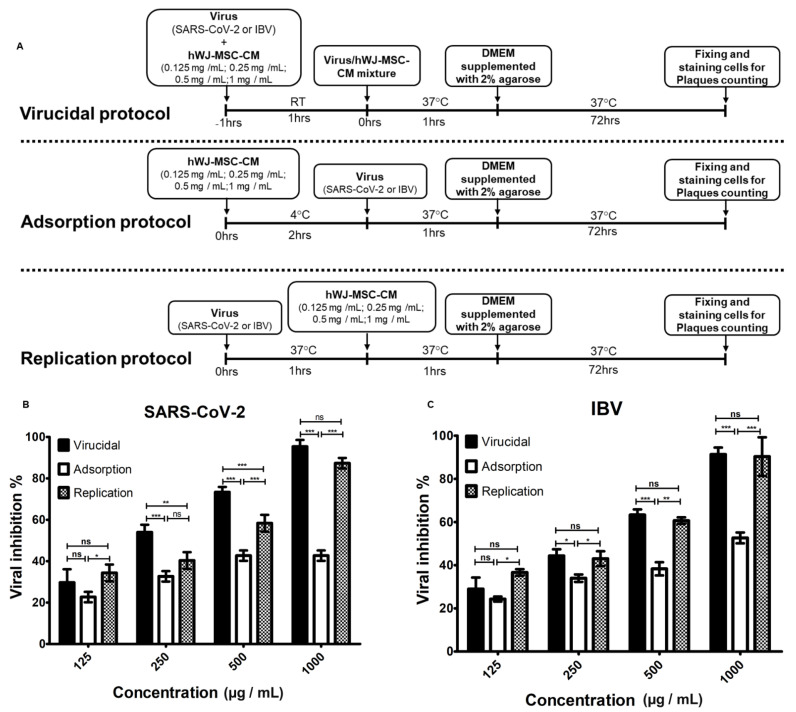
Antiviral mechanism of hWJ-MSC-S. (**A**) Schematic depicting the experimental protocols used for the assessment of antiviral mechanisms of hWJ-MSC-S. Virucidal, anti-adsorption, and anti-replication activities of hWJ-MSC-S against SARS-CoV-2 (**B**) and IBV (**C**) were evaluated in Vero E6 cells and measured by plaque reduction assay. Results are shown as means ± SD of three independent experiments, each run in triplicate. The *p* value; * *p* < 0.05, ** *p* < 0.01, *** *p* < 0.001 indicated the significant correlation among antiviral assays.

**Figure 4 cells-11-01408-f004:**
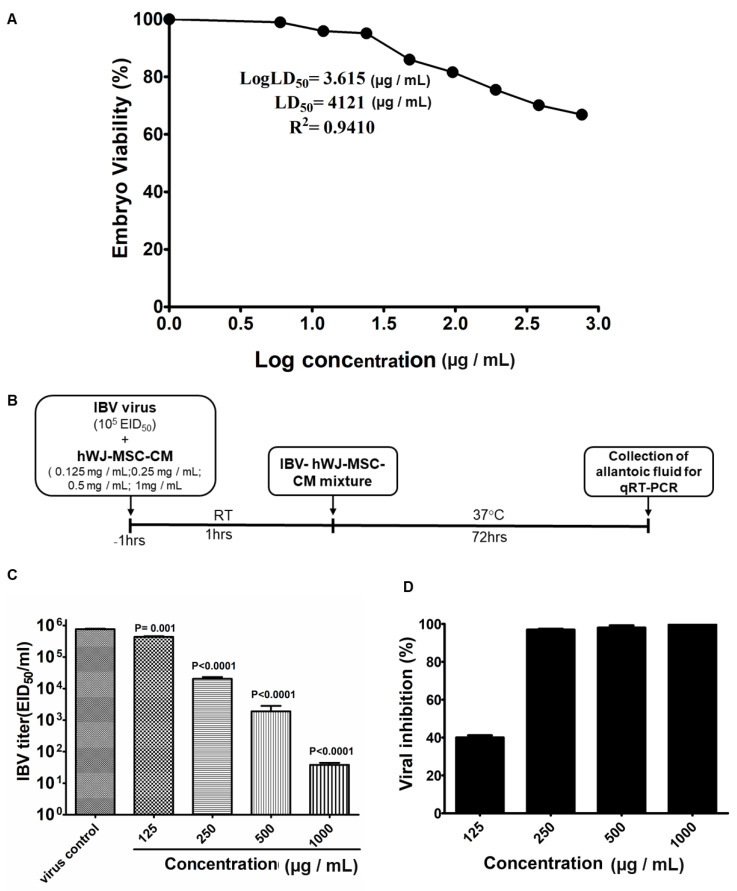
*In ovo* toxicity and anti-IBV activity of hWJ-MSC-S. (**A**) Lethal Dose 50 (LD_50_) of hWJ-MSC-S in SPF-ECEs. The LD_50_ was assessed by inoculating SPF-ECEs with different hWJ-MSC-S concentrations and embryo viability was checked daily. The LD_50_ was calculated as the concentration that causes the death of 50% of inoculated embryos. (**B**) Schematic depicting the experimental protocols used for the assessment of *in ovo* toxicity and anti-IBV activity of hWJ-MSC-S. (**C**) Reduction in viral titer after treatment of IBV with different concentrations of hWJ-MSC-S (125, 250, 500, 1000 µg/mL). (**D**) Percentages of IBV inhibition after treatment with different concentrations of hWJ-MSC-S (125, 250, 500, 1000 µg/mL). Results are shown as means ± SD of three independent experiments, each run in triplicate.

**Figure 5 cells-11-01408-f005:**
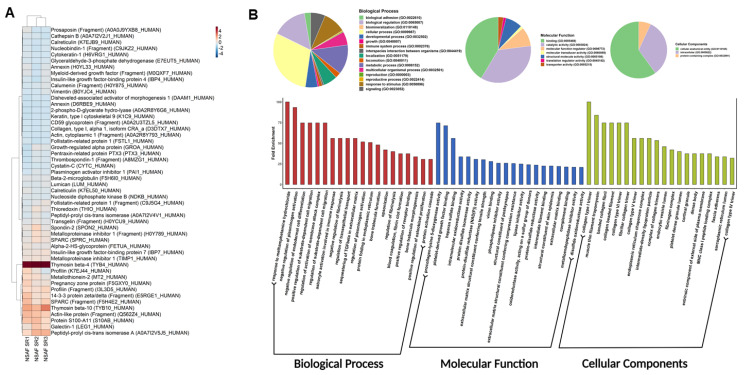
(**A**) Heatmap for the top 50 NSAF proteins for the three replicates. Clustered by Ward algorithm and scaled by each replicate. Associated scale represents scales values of NSAF metric. (**B**) Gene ontology analysis. The upper side (pie chart) of the figure shows the biological process, molecular function, and cellular component total protein percentage hits in each analysis. The bar plot shows the top 20 hits of enrichment biological process (red), molecular function (blue), and cellular component (yellow). Y axis represents enriched fold change.

**Figure 6 cells-11-01408-f006:**
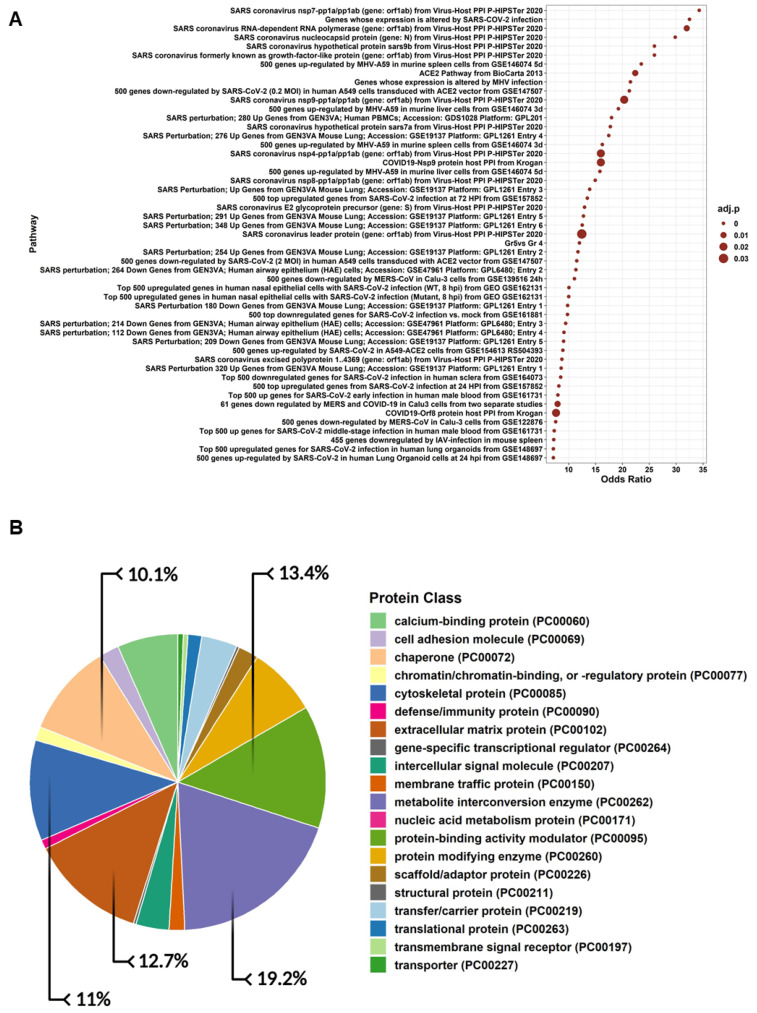
(**A**) Pathway enrichment analysis illustrates the top 10 odds ratio pathways against the COVID-19 database. The pathway was sorted by odds ratio from larger to smaller ones. The size of each point is proportional to the adjusted *p*-value (adjusted *p*-value < 0.05). (**B**) The pie chart represents the percentage of proteins involved in specific protein classes.

**Table 1 cells-11-01408-t001:** CC_50_, IC_50_, and SI of hWJ-MSC-S.

Virus	Cell	CC_50_ (µg/mL)	IC_50_ (µg/mL)	SI (CC_50_/IC_50_)
SARS-CoV-2	Vero-E6	39,200	166.6	235.29
IBV	Vero-E6	3900	439.9	8.87

CC_50_, 50% cytotoxic concentration; IC_50,_ 50% inhibitory concentration; SI, Selective index (CC_50_/IC_50_).

## Data Availability

The mass spectrometry proteomics data have been deposited to the ProteomeXchange Consortium via the PRIDE [96] partner repository with the dataset identifier PXD030966 and 10.6019/PXD030966.

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
