# Peer review of "Human Wharton’s Jelly Mesenchymal Stem Cells Secretome Inhibits Human SARS-CoV-2 and Avian Infectious Bronchitis Coronaviruses"

_cells, 2022, doi:10.3390/cells11091408_

Round 1

Reviewer 1 Report

Reviewer's comments

The manuscript “ Human Wharton’s jelly mesenchymal stem cells secretome inhibits human SARS-CoV2 and avian infectious bronchitis coronavirus” describes a well-elaborated study that looked into the characterization and various aspects of the antiviral activity of hWJ-MSC-secretome. Well depicted figures and illustrations provide the reader clear understanding of the results and methods. The data generated by conducting numerous assays were used to support the drawn conclusion in the manuscript. Here is a list of issues that should be addressed to improve the paper:

Major comments

Line 52: IBV affects avian species, specifically chickens. Poultry production is a big industry worldwide involving a massive number of chickens. How does the author envision the practical use of hWJ-MSC-S against IBV infection compared to SARS-Cov2?

Figure 3: Why was the statistical analysis done between each assay? What was the collected information from these comparisons? Comparison between different concentrations for each assay would provide some info rather than comparing assays for each dose.

Table 1: Please check the SI index (CC50/IC50) for IBV. It should be 8.87 (3900/439.9).

Figure 1 & Line 132-142: The figure shows CD73 & CD34 expression. However, these were not mentioned in the method section. Was the live/dead stain used to exclude dead cells during flow cytometry? What was the use of CD146 in the staining protocol, as this was not indicated in the results/figures?

Extensive editing of text should be considered (grammar, spelling, repetitive wording, variation of font size in a word, the numbering of references, etc)

Minor comments

Line 45: Check the spelling of “ Virucidal”

Line 47: Please spell “LC/MS-MS” out when using it first time.

Line 63: A causative agent is SARS-Cov2, “infection” is inappropriate in the sentence.

Line 144: Should be 80% confluency

Line 301: CD90 was mentioned twice instead of CD73

Line 314: Correct duplication of word “ different”

Line 369: It seems that the statistical comparison was made between each assay, as shown in the figure. However, it was mentioned that the comparison was made against an untreated control and provides the significance of p values. Please make it clear.

Figure 6: Figures A and B are not correctly indicated in the figure legend. Please change them appropriately.

Reviewer 2 Report

#1 The authors present a well-written manuscript, and the context is very helpful: they present new and effective approaches to controll the SARS-CoV-2 pandemic.

#2 The title is informative, short and reflects the content of the study and the abstract reflects the content of the manuscript. However, I strongly suggested that the authors must change the subfamily Coronavirinae to Orthocoronavirinae.

#3 The literature used is appropriate and clarify the authors findings or conclusions.
